# Onward Spread from Liver Metastases Is a Major Cause of Multi-Organ Metastasis in a Mouse Model of Metastatic Colon Cancer

**DOI:** 10.3390/cancers16051073

**Published:** 2024-03-06

**Authors:** Liza A. Wijler, Bastiaan J. Viergever, Esther Strating, Susanne J. van Schelven, Susanna Poghosyan, Nicola C. Frenkel, Hedy te Rietmole, Andre Verheem, Danielle A. E. Raats, Inne H. M. Borel Rinkes, Jeroen Hagendoorn, Onno Kranenburg

**Affiliations:** 1Laboratory of Translational Oncology, Division of Imaging and Cancer, University Medical Centre Utrecht, Heidelberglaan 100, 3584 CX Utrecht, The Netherlands; l.a.wijler-2@umcutrecht.nl (L.A.W.); b.j.viergever-3@umcutrecht.nl (B.J.V.); e.a.strating@umcutrecht.nl (E.S.); s.j.vanschelven-2@umcutrecht.nl (S.J.v.S.); s.poghosyan-2@umcutrecht.nl (S.P.); n.c.frenkel-3@umcutrecht.nl (N.C.F.); hedyterietmole@gmail.com (H.t.R.); a.verheem@umcutrecht.nl (A.V.); d.a.e.raats@umcutrecht.nl (D.A.E.R.); i.h.m.borelrinkes@umcutrecht.nl (I.H.M.B.R.); j.hagendoorn-3@umcutrecht.nl (J.H.); 2Department of Surgical Oncology, Division of Imaging and Cancer, University Medical Centre Utrecht, Heidelberglaan 100, 3584 CX Utrecht, The Netherlands; 3Utrecht Platform for Organoid Technology, Utrecht University, Heidelberglaan 8, 3584 CS Utrecht, The Netherlands

**Keywords:** colorectal cancer, liver metastases, onward metastasis, angiogenesis, macrophages, vitronectin

## Abstract

**Simple Summary:**

The prognosis of patients with metastatic colon cancer remains very poor. A minority of patients, mostly those with liver-restricted disease, are eligible for intentionally curative surgery. In most cases, however, the presence of extra-hepatic metastases precludes curative treatment. The aim of this study was to gain insight into the processes governing multi-organ metastasis. We applied the microsurgical transplantation of mouse colon tumor organoids into the caecum or into the liver of syngeneic immunocompetent mice. Colon tumors growing in the liver seeded distant metastases to the lungs and to the peritoneal cavity more efficiently than those growing in the caecum. This was associated with the formation of hotspots of macrophage-surrounded vitronectin-positive blood vessels, specifically in liver tumors. Thus, ‘onward spread’ from liver metastases plays a major role in multi-organ metastasis, potentially through liver-specific vascular hotspots. The therapeutic targeting of these signals may help achieve the containment of the disease within the liver, thus preventing multi-organ metastasis.

**Abstract:**

Colorectal cancer metastasizes predominantly to the liver but also to the lungs and the peritoneum. The presence of extra-hepatic metastases limits curative (surgical) treatment options and is associated with very poor survival. The mechanisms governing multi-organ metastasis formation are incompletely understood. Here, we tested the hypothesis that the site of tumor growth influences extra-hepatic metastasis formation. To this end, we implanted murine colon cancer organoids into the primary tumor site (i.e., the caecum) and into the primary metastasis site (i.e., the liver) in immunocompetent mice. The organoid-initiated liver tumors were significantly more efficient in seeding distant metastases compared to tumors of the same origin growing in the caecum (intra-hepatic: 51 vs. 40%, *p* = 0.001; peritoneal cavity: 51% vs. 33%, *p* = 0.001; lungs: 30% vs. 7%, *p* = 0.017). The enhanced metastatic capacity of the liver tumors was associated with the formation of ‘hotspots’ of vitronectin-positive blood vessels surrounded by macrophages. RNA sequencing analysis of clinical samples showed a high expression of vitronectin in liver metastases, along with signatures reflecting hypoxia, angiogenesis, coagulation, and macrophages. We conclude that ‘onward spread’ from liver metastases is facilitated by liver-specific microenvironmental signals that cause the formation of macrophage-associated vascular hotspots. The therapeutic targeting of these signals may help to contain the disease within the liver and prevent onward spread.

## 1. Introduction

Colon cancer is the third most frequently diagnosed type of cancer and the second leading cause of cancer-related mortality worldwide [1]. Patients diagnosed with metastatic colon cancer have a very poor prognosis, with a 5-year survival chance of ~10–15%. The liver is the most common site of distant metastasis formation (~75%), presumably because venous blood from the colon directly enters the liver via the portal vasculature. The most common extra-hepatic metastatic sites are the lungs and the peritoneal cavity [2]. 

If metastatic colon cancer is confined to the liver, surgical resection of the liver metastases represents a potentially curative treatment strategy. However, the presence of extra-hepatic metastases is usually a contra-indication for such treatment. For patients with mCRC who are not eligible for liver surgery, systemic therapy or other forms of palliative treatment are available [3]. 

The prevailing model of liver metastasis formation in CRC describes a stepwise process in which a fraction of tumor cells within the primary tumor acquires specific characteristics that allow them to break away from their neighboring cells and gain access to the vasculature. These disseminated cells are then transported via the inferior mesenteric vein and the portal vein into the liver, where they arrest in the microvasculature [4,5]. Following extravasation, some cells may resume growth at sites of preformed pre-metastatic niches [6,7]. These micro-metastases may grow into macro-metastases following an ‘angiogenic switch’ and successful immune evasion [8]. This model, however, does not describe the consequences of LM formation. This is relevant because the mechanisms that lead to the formation of extra-hepatic metastases are incompletely understood. An important question is whether distant metastases are predominantly seeded from primary tumor-derived cells or whether ‘primary’ metastases can seed ‘secondary’ metastases in other liver segments and/or additional distant sites. 

A comprehensive understanding of multi-organ metastasis is likely to influence its clinical management in multiple ways. For instance, the presence of metastases in multiple organs is associated with poor survival and often precludes liver surgery, thereby greatly limiting curative treatment options [2]. Mechanistic insight into the formation of extra-hepatic metastases may provide a basis for developing predictive biomarkers and effective therapeutic strategies that aim to contain the disease within the liver, thus increasing operability and improving survival. 

Analyses of driver gene mutations in different metastases from individual (untreated) patients have revealed that these metastases are remarkably similar [9]. This suggests that a single (sub-) clone within the primary tumor seeds the vast majority of distant metastases. While these analyses provide insight into the genetic relationships among metastases, they do not provide information regarding the anatomical origin of the seeding of ‘common ancestor’ cells. A complete understanding of multi-organ metastasis therefore requires genetic data describing the evolutionary relationships and heterogeneity within and between metastases and the primary tumor, as well as insight into the dominant anatomical routes that give rise to metastases growing at distinct sites. 

Here, we tested the hypothesis that metastatic capacity is influenced by the site of tumor growth. To this end, we transplanted murine colon cancer organoids into the caecum (the primary tumor site) or into the liver (the primary metastasis site) of immune-competent mice and compared the extent and patterns of spontaneous metastasis formation. Moreover, our analysis of RNA sequencing data of primary tumors and liver metastases derived from cancer patients provided insight into site-specific patterns of gene expression potentially influencing metastatic capacity. The results from both experimental approaches show that a liver-specific program fostering the generation of angiogenic hotspots may facilitate onward (secondary) spread from liver metastases.

## 2. Materials and Methods

### 2.1. Organoid Culture

Organoids were derived as described in [10] from spontaneously formed primary tumors (MTDO2-4) and a liver metastasis (MTDO1) in a transgenic mouse model with the expression of the Notch1 intracellular domain and deletion of p53 in the digestive tract [11]. Exome sequencing revealed that all tumors harbor mutations in either the Ctnnb1 or Apc genes, demonstrating classical Wnt pathway activation. CRC organoids were transduced with a lentiviral vector expressing luciferase and dTomato (pUltra-Chili-Luc, Addgene #48688, Watertown, MA, USA) and were FACS-purified. The CRC organoids were cultured in droplets of Growth Factor-Reduced Basement Membrane Extract (BME; Amsbio, Cambridge, MA, USA) and Advanced DMEM/F12 (Thermo Fisher Scientific, Waltham, MA, USA) supplemented with 1% Penicillin–Streptomycin (Gibco, Grand Island, NY, USA), 1% HEPES buffer, 2 mM Glutamax (Invitrogen, Waltham, MA, USA), 2% B27 supplement (Invitrogen), 100 ng/mL Noggin (produced by lentiviral transfection), 10 nM murine recombinant FGF (PeproTech, London, UK), and 1 mM n-Acetylcysteine (Sigma-Aldrich, St. Louis, MO, USA). The CRC organoids were passaged using TrypLE weekly, and medium was refreshed twice a week. CRC organoid cultures were maintained at 37 °C in a humidified atmosphere containing 5% CO_2_. 

### 2.2. Ethical Guidelines

This study was conducted in accordance with institutional guidelines for the care and use of laboratory animals. All animal procedures related to the purpose of the research were approved by the Animal Welfare Body under the Ethical license of University Utrecht, Medical Center Utrecht, The Netherlands, as filed by the relevant national authority, ensuring full compliance with the European Directive 2010/63/EU for the use of animals for scientific purposes.

### 2.3. Animals

Male C57BL/6NCrl mice, 8–10 weeks of age, were group-housed in open cages with contact bedding, plastic enrichment shelter, and tissues for nesting. Upon arrival in the facility, the mice were housed for at least one week before tumor inoculation was performed. The mice were fed ad libitum with AIN-93M pellets (Ssniff Spezialdiäten GmbH, Soest, Germany), had unlimited access to water, and were maintained on a 12:12 h light (7 AM)/dark (7 PM) cycle at 20–24 °C, 45–60% humidity. Body weight was monitored weekly, and tumor progression was monitored by monitoring clinical signs, bioluminescent imaging (weekly), or, for subcutaneous models, by caliper measurements (biweekly). 

### 2.4. Subcutaneous Tumor Initiation

For subcutaneous injections, tumor organoids were disassociated using TrypLE (Gibco) and resuspended in Advanced DMEM/F12 medium. Prior to injection, cold BME was added to the organoid suspension (250.000 single cells) in a ratio of 1:1. After gently mixing by inversion, 100 µL organoid/BME suspension was injected per inoculation site (both flanks).

### 2.5. Microsurgical Implantation of Tumor Organoids into the Caecum and the Liver

For orthotopic models, tumor organoids were implanted in the liver and caecum, respectively (Appendix A). Prior to surgery, the mice were anesthetized using isoflurane 3–5% for induction and 1.5–2% for maintenance and were administered the analgesic buprenorphine (0.1 mg/kg) subcutaneously. The CRC tumor organoids were disassociated by TrypLE (Gibco); 250,000 single cells were embedded in 6 µL of 75% Rat Tail Type I Collagen (Corning, New York, NY, USA) droplets and 25% neutralization buffer (AlphaMEM powder, Life Technology, Carlsbad, CA, USA), 1 M HEPES buffer pH 7.5 (Invitrogen), and NaHCO_3_ (Sigma). After overnight recovery, collagen droplets containing 1-day-old CRC organoids were implanted into the median liver lobe using microsurgery techniques. Using microsurgical equipment and light microscopy, an incision of 0.5–1 cm was made in the skin, followed by the opening of the peritoneum wall. A sterile cotton tip was used to position the liver or the caecum onto a sterile gauze outside the abdomen, where a 2 mm incision was made in the median liver lobe or the serosa of the caecum using a scalpel. To prevent excessive bleeding, a cotton tip was placed on the incision until an air-dried collagen droplet was gently pushed into the incision and sealed with Seprafilm (Genzyme, Cambridge, MA, USA). The peritoneum wall and skin were sutured. The mice were closely monitored during recovery and two days after to ensure the proper closure of the incision wounds. The mice were sacrificed under anesthesia by cervical dislocation upon showing signs of tumor growth and metastasis formation (clinically and by BLI). All relevant tumor-bearing organs and blood were harvested for further analyses.

### 2.6. Bioluminescence Imaging 

Tumor progression and the formation of metastases were monitored in orthotopic models using in vivo bioluminescence imaging (BLI). The mice were anesthetized (isoflurane 4% for induction and 2% for maintenance, with 1.6 L/min oxygen) and received an intraperitoneal injection with 100 μL of D-luciferin in PBS. The mice were imaged for 5 min (1 s exposure/image) using the PhotonIMAGERTM RT system (Biospace Lab, Paris, France). The images were analyzed using M3 Vision software version 2.2.1 (Biospace Lab). The number of photons per minute within a region of interest was recorded and expressed as counts per second (cps). After sacrifice, the organs of each mouse were individually imaged ex vivo to assess the quantity of tumors per site of metastasis.

### 2.7. Immunohistochemistry

Tumor-bearing tissues were fixed in formalin and embedded in paraffin (FFPE). Next, 4 µm FFPE slides were cut, antigen retrieval was performed using citrate buffer, and endogenous peroxidase was blocked for 30 min. A detailed description of the staining protocols can be found in Appendix A. The antibodies we used are as follows: anti-EpCAM (#50591-R002, 1:1000, Sino Biological, Wayne, PA, USA), anti-Vimentin (#5741S, 1:100, Cell Signaling, Danvers, MA, USA), anti-α-Smooth Muscle Actin (#AB5694, 1:200, Abcam, Cambridge, UK), anti-vitronectin (PA5-27909, 1:1000, Thermofisher), and BrightVision Poly-HRP-Anti Rabbit Biotin-free (Immunologic, ready-to-use), followed by 3,3′-diaminobenzidine tetrahydrochloride (DAB) incubation and counterstaining with Mayer’s Hematoxylin and Eosin (HE). Collagen type I and III was visualized using Picro Sirius Red staining (Sigma Aldrich). Finally, the slides were mounted with coverslips using ClearVue™ Coverslipper (ThermoFisher). The slides were digitally scanned and loaded in QuPath software (v0.2.3), and hotspot areas were visually identified based on high CD31 expression, with distinctive, open vasculature. The CD31 and CD68 stainings were quantified in QuPath by using the cell detection tool, selecting positive cells based on a predetermined cut-off value to quantify DAB-positive cells in total tumor and hotspot areas. Vitronectin positivity (%) was quantified by counting the positive vessels per total identified intra-tumor vessel.

### 2.8. Fluorescent Multiplex Immunohistochemistry

For the simultaneous detection of EpCAM, α-SMA, and PDGFR-α (3164S, 1:200, Cell signaling), fluorescent multiplex IHC involving the tyramide signal amplification methodology was performed using Tyramide SuperBoost™ Kits Alexa Fluor (AF)488, AF555, and AF647 (ThermoFisher). Epitope retrieval was performed in 10 mmol/L sodium citrate (pH 6.0) or 1 mmol/L EDTA (pH 9.0), endogenous peroxidase was inactivated, and the slides were blocked with 10% goat serum prior to primary antibody incubation. The primary antibody was incubated overnight (4 °C) before incubation with AF-conjugated tyramides. Subsequently, same-species primary antibodies were applied after 10 min of the heat-mediated stripping of the antibody complex in citrate buffer and repeating the same procedure above. The slides were counterstained with DAPI and mounted with ProLong^®^ Gold Antifade Mountant (ThermoFisher) and a coverslip before imaging using confocal microscopy (LSM 510 META). 

### 2.9. DNA Isolation and Library Preparation

DNA was isolated using the QIAamp^®^ DNA mini kit according to the manufacturer’s instructions. A total of 500–1000 ng of DNA per sample was used for DNA library preparation. Library preparation was performed using the Truseq DNA nano protocol. 

### 2.10. Whole Genome Sequencing 

Whole genome sequencing was performed by the Hartwig Medical Foundation using the Illumina NovaSeq 6000 set-up (Illumina, San Diego, CA, USA) and further analyzed by the Utrecht Sequencing Facility (USEQ) using the reference genome assembly GRCm38. Paired-end whole genome sequencing was performed with an average coverage of 30×. Mapping and variant calling were performed using HaplotypeCaller (v3.4). Variant copy frequency files were filtered based on the quality of variants (>100.0) using the Galaxy online analysis platform [12]. 

### 2.11. Statistical Analyses

All data are represented as Mean ± SD. Statistical analyses were performed using SPSS (Version 27). Statistical differences were considered significantly different if *p* < 0.05. A generalized estimating equations (GEE) model was used to compare differential metastasis formation of MTDO4 between the liver (*n* = 44) and caecum implantation (*n* = 16) models. For the immunohistochemistry data, a non-parametric *t*-test was performed to compare two groups. Illustrations were created using Biorender (Biorender.com (accessed on date 26 June 2023)); figures were created using Prism Graphpad (Version 8) and Adobe Illustrator (25.4.1).

## 3. Results

### 3.1. Generation of Organoids for Modeling Metastatic CRC in Immunocompetent Mice 

We used a transgenic mouse model in which the expression of the Notch1 Intracellular Domain (NICD) and deletion of *TP53* in the digestive epithelium drive the formation of metastatic intestinal cancer [11]. Within a period of 15 months, these mice develop invasive adenocarcinomas with reactive stroma (100%), peritoneal metastases (50%), and liver metastases (10%). To increase the robustness and speed of the model, we generated tumor organoids from three distinct primary tumors (MTDO2-4) and from one liver metastasis (MTDO1) for transplantation purposes (Figure 1A). Whole genome sequencing of these mouse tumor-derived organoids (MTDOs) confirmed the presence of classical colorectal tumor Apc mutations and SMAD-4 stop and frameshifting mutations (Appendix A). When injected subcutaneously into the mice, all four organoids were able to initiate tumor formation. Histological and immunohistochemistry analysis showed that the tumors displayed multiple features of aggressive behavior, including a high amount of reactive stroma (Figure 1B) and the presence of budding individual tumor cells and clusters into the tumor stroma (Figure 1C), similar to the spontaneous tumors that were described in the original report [11].

### 3.2. Metastatic Capacity Is Influenced by Tumor Location

The generated MTDOs were then used to model spontaneous metastasis formation by tumors growing in the intestine (the primary tumor site) versus those growing in the liver (the primary metastasis site) of immunocompetent C57/Bl6 mice. All MTDOs were transduced with a lentiviral vector driving the expression of firefly luciferase, which allowed for non-invasive bioluminescence imaging (BLI). Luciferase-expressing organoids were then embedded in collagen and transplanted either into the caecum part of the large intestine or into the liver (Figure 2A,B; Appendix A). This resulted in the formation of a single tumor at both implantation sites in all mice (100% versus 100%). Caecum and liver tumors at the site of implantation were first detectable by BLI within 2 weeks (Figure 2C). The BLI measurements demonstrated metastatic spread of the disease in the liver-implanted mice. All mice were sacrificed at signs of discomfort (humane endpoint). This revealed that survival was significantly reduced in the mice with liver tumors compared to the mice with caecum tumors (Figure 2D). 

To quantify the extent of metastasis in both models, we performed post mortem BLI measurements (Figure 2C and Figure 3A), a visual inspection (Figure 3A), and histological tissue analyses (Figure 3B). This created a binary score per tumor/metastasis location (1: tumor present; 0: no tumor present). The data of four independent experiments were then pooled and analyzed using a population-based GEE model (Appendix A). These analyses revealed that liver tumors seed liver metastases (to contralateral liver lobes) significantly more efficiently than caecum tumors (51% versus 40%; *p* = 0.001) (Table 1). Likewise, liver tumors were also significantly more efficient in seeding peritoneal metastases (51% versus 33% *p* = 0.001) and lung metastases (30% versus 7%; *p* = 0.017) when compared to caecum tumors (Figure 3C and Table 1).

### 3.3. Macrophage-Associated Vascular Hotspots in Liver Metastases: Potential Portals for Onward Spread

RNA sequencing of clinical tissue samples derived from liver metastases and primary tumors [13] revealed that liver metastases express significantly higher levels of signatures reflecting tissue hypoxia and hypoxia-induced vascular endothelial growth factor A (VEGFA) (Figure 4A). A previous work showed that liver metastases from CRC are hypoxic and that this is associated with macrophage infiltration [14]. Indeed, macrophage markers CD68 and CD163 are expressed at significantly higher levels in liver metastases than in paired primary tumors (Figure 4B). Differential gene expression analysis further showed that the extracellular matrix component vitronectin (VTN) is highly expressed in liver metastases but not in primary colon tumors (Figure 4C), in line with previous findings [15]. Additionally, combined expression analysis of VEGFA/CD68 and VTN/CD163 largely separated primary tumors from liver metastases (Figure 4D). A further analysis of immune cell signature expression [16] showed that signatures reflecting the presence of B cells and T helper cells are expressed at significantly lower levels in liver metastases than in primary tumors (Appendix A). Vice versa, liver metastases expressed higher levels of the metabolic Hallmark signatures [17] ‘bile acid metabolism’ and ‘xenobiotic metabolism’ (Appendix A). Other Hallmark signatures were not significantly differentially expressed. 

To assess how the mouse model for metastatic colon cancer described in this report reflects these site-specific differences in the tumor microenvironment, we performed immunohistochemistry analysis to detect blood vessels (CD31), macrophages (CD68), and vitronectin. In line with the clinical data (Figure 4), the infiltration of CD68^+^ macrophages was significantly higher in the liver tumors when compared to the primary tumors (Figure 5A). The macrophages were clustered in ‘hotspots’ surrounding intra-tumor CD31^+^ blood vessels (Figure 5B). Vitronectin expression was primarily found in macrophage-associated vascular hotspots in the liver tumors but not in the primary tumors (Figure 5C). 

### 3.4. Expression of a Liver Metastasis Signature in Primary CRC Predicts Distant Metastasis Formation

Previous work has identified a signature distinguishing liver metastases from paired primary tumors [13]. Our data suggest that such a signature may identify metastasis-prone tumors. To test this, we generated liver metastasis signature scores in a large cohort of primary colon cancer with relapse-free survival data. Primary tumors with a high expression of the liver metastasis signature were significantly more prone to develop distant metastases than those expressing intermediate or low levels of the signature (Figure 6A). Furthermore, the tumors with a high expression of the liver metastasis signature consisted almost entirely of the mesenchymal, stroma-rich, Consensus Molecular Subtype 4 (CMS4) (Figure 6B) [19]. In line with these findings, the liver metastasis signature was expressed at the highest levels in CMS4 tumors compared to CMS1-3 tumors (Figure 6C). Expression of the liver metastasis signature correlated extremely well with expression of the CMS4-identifying genes from the original random forest CMS classifier (r = 0.89; Figure 6D). Interestingly, despite the strong correlation of the liver metastasis signature with CMS4, the expression of stromal genes (FAP, smooth muscle actin (ACTA2)) and the ESTIMATE tumor stroma signature were not significantly different between liver metastases and paired primary tumors (Figure 6E–G), while such genes are significantly more expressed in CMS4 tumors compared to CMS1-3 tumors [19,20]. Thus, specific aspects of the aggressive CMS4 phenotype (VTN+ blood vessels, macrophage infiltration) are more pronounced in liver metastases when compared to paired primary tumors, but these features are unrelated to stromal content.

## 4. Discussion

In this report, we present a novel immunocompetent mouse model for spontaneous onward colon cancer metastasis initiated by liver metastases. The onward spread of tumor cells originating from liver metastases is a highly relevant phase of the metastatic process, as it determines the extent of intra- and extra-hepatic metastasis formation. Comparing metastatic spread from primary tumor versus onward metastases from liver tumors, we found the latter to be considerably more efficient and profound. Interestingly, this increased metastatic capacity of liver metastases compared to primary tumors was not accompanied by a significantly higher stromal content, one of the most distinguishing features of the metastasis-prone primary colon cancer subtype CMS4 [19]. This suggests that a high stromal content per se is unlikely to explain the aggressive onward metastasis-prone behavior of liver metastases. Rather, we found that other aspects of the CMS4 phenotype play an important role in onward metastasis from the liver. Macrophages have been extensively studied for their role in cancer progression and metastasis. Tumor-associated macrophages (TAMs) can promote tumor angiogenesis, matrix remodeling, and tumor cell motility, thereby contributing to metastasis. CD163^+^ TAMs induce the epithelial-to-mesenchymal transition (EMT) and thereby enhance colorectal cancer cell migration and invasion [21]. TAMs help tumor cells enter blood vessels, thereby supporting tumor cell dissemination and metastasis [22,23,24,25]. 

Aside from an enrichment of macrophages in liver metastases compared to primary tumors, we also found a significantly lower expression of signatures reflecting the presence of B cells and T helper cells in liver metastases compared to primary tumors. Interestingly, the presence of liver metastases is associated with a poor response to immune checkpoint inhibitors [26]. This is due to liver-resident macrophages that produce FAS ligands to initiate apoptosis in FAS-expressing activated T cells [26]. The mice in our model system are immune-competent. Therefore, the efficient seeding of secondary metastases by liver metastases may also be (in part) the result of the macrophage-induced inhibition of anti-tumor immunity. 

Liver metastases express significantly higher levels of the signatures ‘bile acid metabolism’ and ‘xenobiotic metabolism’. This is, in all likelihood, a reflection of the liver microenvironment, in which both metabolic pathways are highly active. 

In this study, we found that the presence of vitronectin-positive blood vessels surrounded by tumor-infiltrating macrophages may play an important role in facilitating the onward spread of tumor cells towards other distant metastatic sites such as the lungs and peritoneum. Therapeutic strategies that effectively target this last phase of liver metastasis formation may help to contain the disease within the liver and thereby generate clinical benefits. To develop such strategies, the key players driving onward spread should be identified. In addition to angiogenesis inhibitors (with proven efficacy), such strategies may involve targeting macrophages, signals promoting macrophage recruitment, and/or vitronectin or other extracellular matrix components making up the liver-specific vascular hotspots. Such therapeutic strategies aimed at containing the disease in the liver and prevent onward spread should therefore be tested in multiple mouse models to provide a robust basis for clinical translation. Moreover, additional research should focus on the ability of metastases growing at other distant sites (i.e., the peritoneal cavity, the lungs, etc.) to seed secondary metastases, as we have studied here for liver metastases. Such efforts may entail the application of mouse models, as well as in-depth comparative analyses of human metastasis samples isolated from distinct organ sites. Importantly, complex processes like multi-organ metastasis are likely to depend on the (epi-)genetic make-up of the original tumors. Therefore, it will be important to study the mechanistic principles of multi-organ metastasis and new therapeutic strategies in multiple genetically distinct models. Organoid technology may be used to generate ‘living biobanks’ of collections of (organ-site specific) metastasis-derived cultures that can subsequently be used in such studies. Ultimately, research using these improved model systems should create a deeper understanding of the mechanisms driving multi-organ metastasis in CRC. The results obtained by applying such advanced models may have a higher chance of successful clinical translation. 

Another important variable that should be considered in future research and CRC modeling is the location of the primary tumor. In particular, tumor cells disseminating from lower rectal tumors may bypass the portal venous system and the liver by entering the systemic circulation via the inferior and middle rectal veins draining into the inferior vena cava [27,28,29]. Indeed, lung metastases occur more frequently in patients with rectal cancer than those with colon cancer [30]. This highlights the importance of also considering the primary tumor site when modeling multi-organ metastasis in CRC. 

Our results may also offer an explanation for why anti-angiogenesis therapy (anti-VEGFA; bevacizumab) generates clinical benefit in the treatment of metastatic colon cancer [31] but not when added to adjuvant chemotherapy following primary tumor resection [32]. 

Together, our data indicate that the formation of vitronectin-positive vascular hotspots with associated macrophages may facilitate the onward spread of tumor cells from liver metastases to other distant sites and thereby enhance multi-organ metastasis.

## 5. Conclusions

We conclude that liver metastases are efficient instigators of multi-organ metastasis and that liver-specific microenvironmental signals are likely to contribute to such ‘onward spread’. Macrophage-associated vascular hotspots, but not a higher amount of tumor stroma per se, are associated with efficient metastatic spread originating from liver tumors. The therapeutic targeting of these tissue-specific signals could help contain the disease in the liver and thereby improve operability and patient survival. Furthermore, comparative RNA sequencing data of clinical tumor samples comparing primary tumors and liver metastases have identified site-specific patterns of gene expression. These data could guide future site-specific therapeutic approaches to combat multi-organ metastasis. The development and testing of such approaches rely on the analysis of clinical tumor samples and on the application of mouse models in which spontaneous multi-organ metastasis is faithfully recapitulated. 

## Figures and Tables

**Figure 1 cancers-16-01073-f001:**
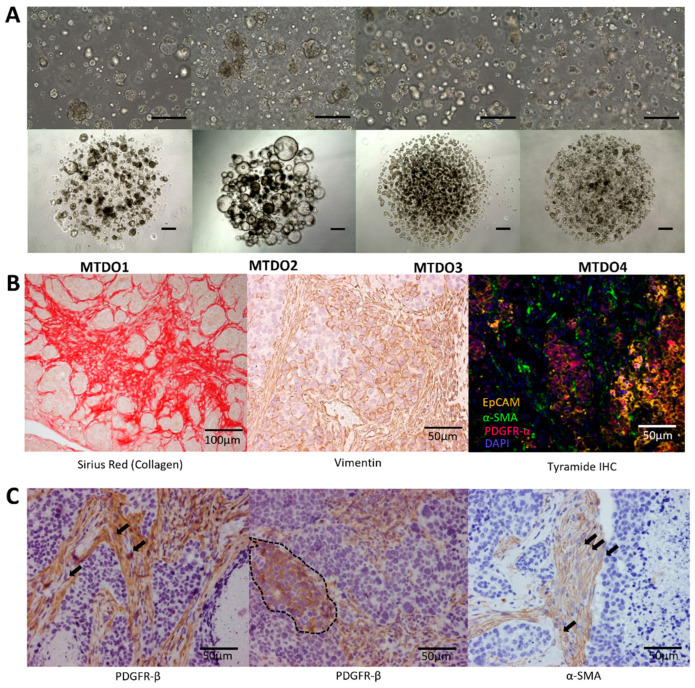
Generation of organoids for modeling spontaneous metastatic CRC in immunocompetent mice. (**A**) Representative brightfield pictures of mouse tumor-derived organoids (MTDO1-4) in 3D BME domes, 10× (top) and 4× (bottom) magnification, scale bars represent 100 µm. (**B**) Histological stainings of Sirius Red (collagen), vimentin, and fluorescent multiplex IHC of subcutaneous tumors; EpCAM depicted in orange, α-smooth muscle actin (SMA) depicted in green, PDGFR-α depicted in red, nuclei (DAPI) depicted in blue. Scale bar represents 100 µm (collagen) or 50 µm (vimentin and multiplex IHC). (**C**) PDGFR-β and α-SMA staining of IHC slides. The arrows indicate the presence of budding tumor cell clusters within the tumor stroma. The dotted line indicates an area of tumor cells interspersed with stromal cells. Scale bar represents 50 µm.

**Figure 2 cancers-16-01073-f002:**
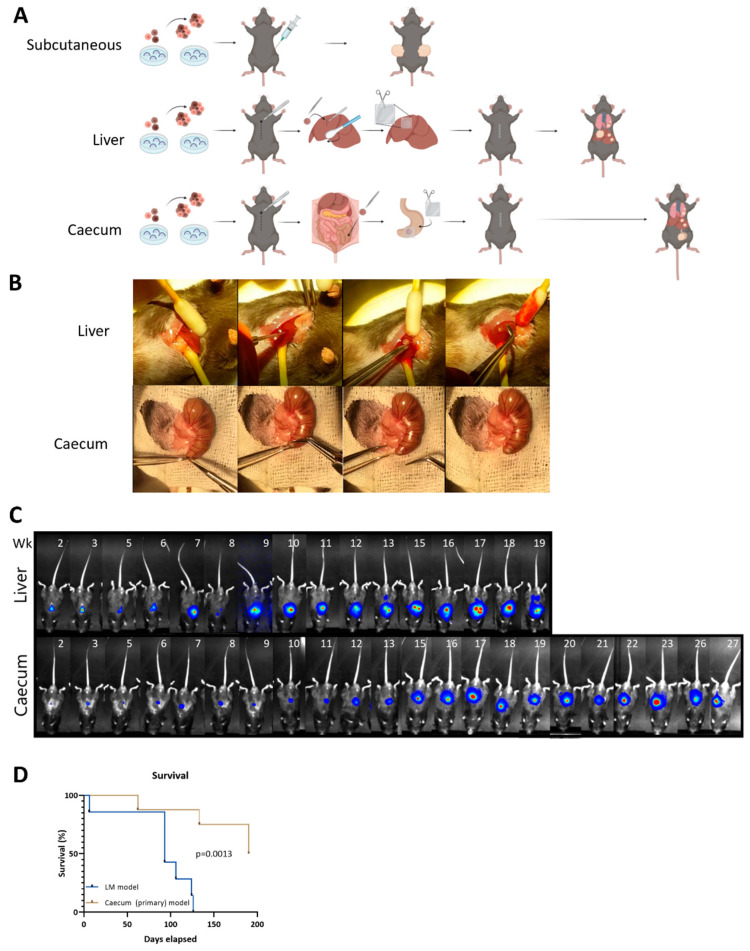
Liver tumors are more aggressive than caecum tumors. (**A**) Schematic representation of subcutaneous liver and caecum (primary) implantation of MTDOs. (**B**) Implantation of MTDO droplets into the liver and caecum. (**C**) Bioluminescent imaging indicated more rapid and extensive metastasis from liver tumors compared to primary caecum tumors. Blue represents low intensity signal, gradual scale to red representing high intensity signal. Numbers indicate weeks following implantation. (**D**) Kaplan–Meier curves comparing liver metastasis (LM) model to primary tumor model using Mantel–Cox test (*p* = 0.0013).

**Figure 3 cancers-16-01073-f003:**
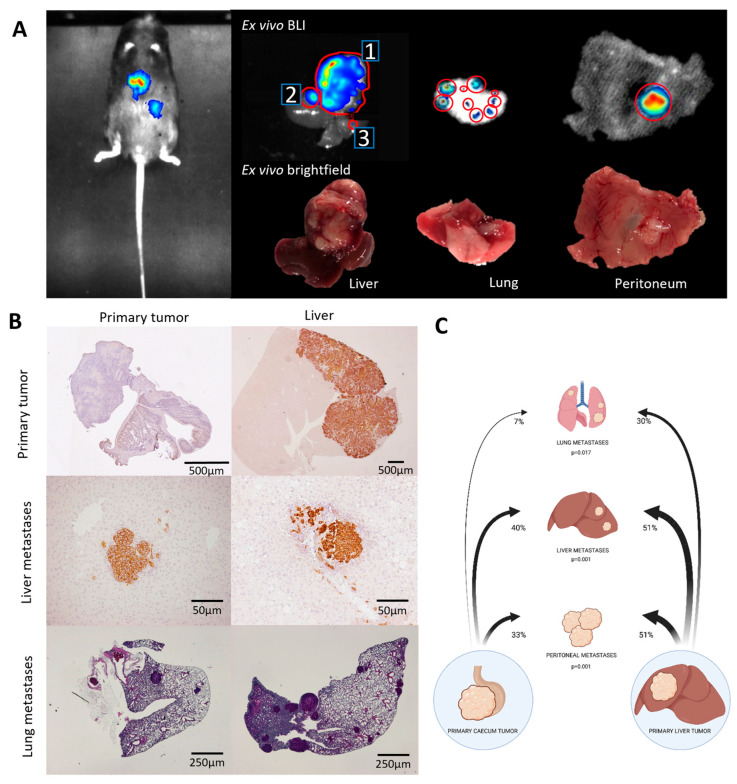
Liver tumors display higher metastatic capacity than caecum tumors. (**A**) BLI at sacrifice—in vivo, post mortem, ex vivo. 1. Primary liver tumor; 2 and 3. Liver metastasis. Red circles: indicating individual tumor lesions. (**B**) Histology of primary tumors and metastases (EpCAM in liver and HE staining in caecum and lung tissue). (**C**) GEE analysis of four independent experiments revealed significantly distinct metastatic efficiency values depending on the primary tumor location.

**Figure 4 cancers-16-01073-f004:**
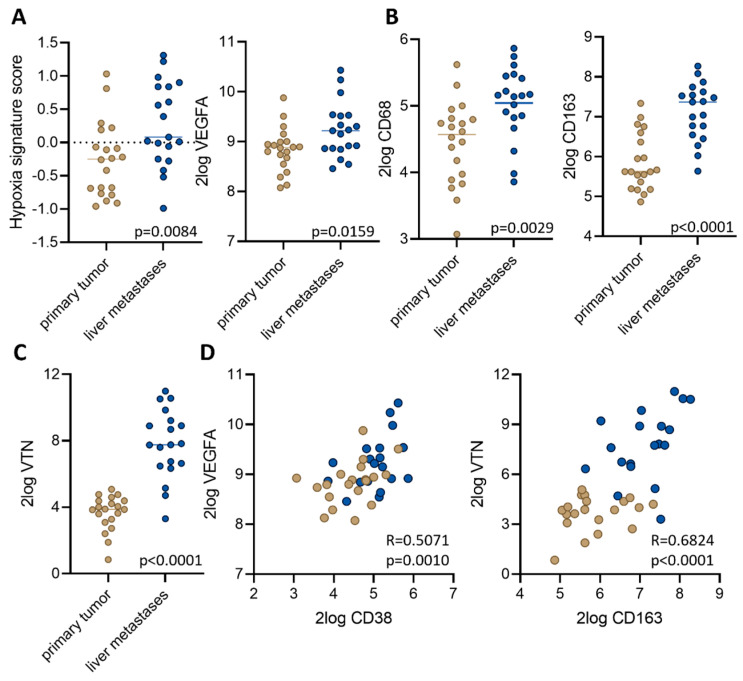
Distinct microenvironments in paired primary tumors and liver metastases. Expression of (**A**) a hypoxia signature [18], (**B**) VEGFA, and (**C**) vitronectin (VTN) in a cohort of paired primary tumors and liver metastases [16]. (**D**) Correlation plots of the expression of CD68 and VEGFA (left panel) or CD163 and VTN (right panel) in the same cohort. Tissue of origin is color coded. Beige: primary tumor samples. Blue: liver metastasis samples.

**Figure 5 cancers-16-01073-f005:**
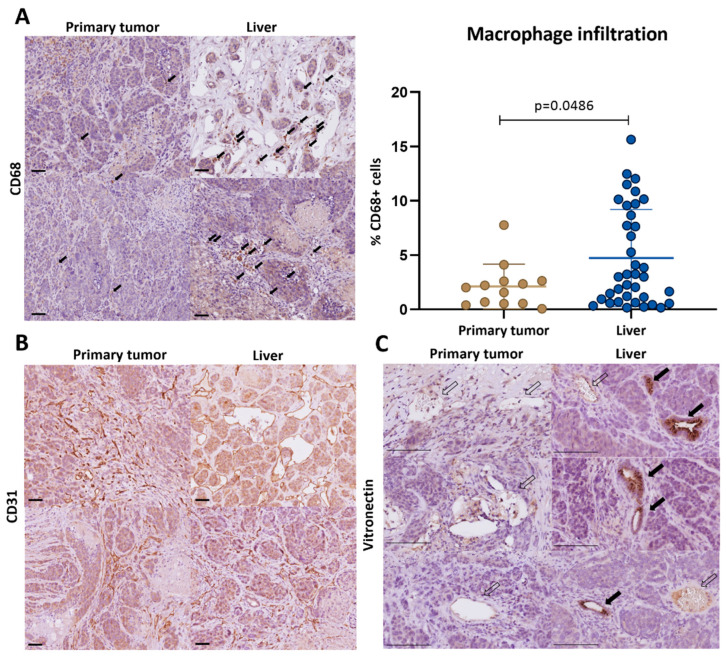
Liver metastases contain macrophage-associated vitronectin-positive vascular hotspots. (**A**) Immunohistochemistry analysis of CD68 in primary tumors and liver metastases. Box plots displaying the quantification of the CD68 signals by QuPath. The infiltration of CD68^+^ macrophages was significantly higher in liver tumors when compared to primary tumors (unpaired *t*-test). Black arrows indicate CD68^+^ macrophages. Scale bars represent 50 µm (**B**). Paired IHC image sections showing clustering of macrophages (CD68) in ‘hotspots’ surrounding intra-tumor CD31^+^ blood vessels. Scale bars represent 50 µm. (**C**) Representative IHC images of vitronectin expression in macrophage-associated vascular hotspots. Black arrows represent vitronectin-positive vessels, white arrows indicate vitronectin-negative vessels. Scale bars represent 100 µm.

**Figure 6 cancers-16-01073-f006:**
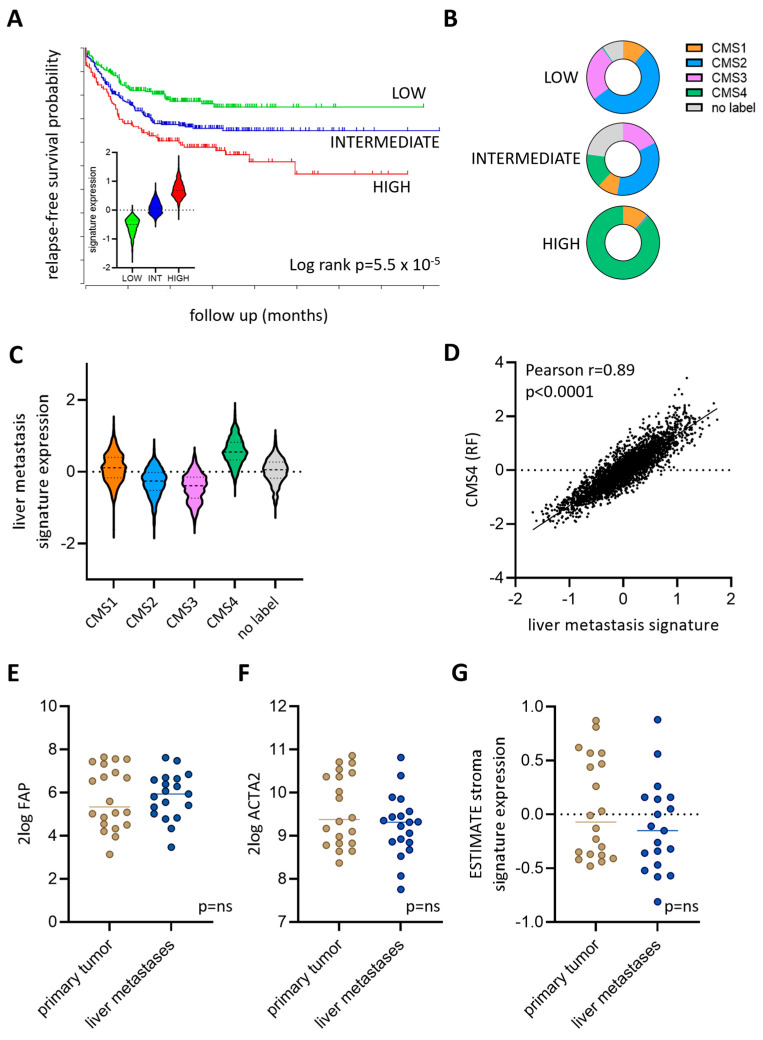
A liver metastasis signature is highly expressed in consensus molecular subtype 4 (CMS4) primary colon tumors. (**A**) The liver metastasis signature was used to cluster the primary colon tumors from a large composite cohort with survival data and annotated CMS classification [19] into expression subgroups. The Kaplan–Meier curves show that the high expression of this signature correlates with an increased likelihood of distant relapse. The violin plot (inset) shows the expression of the signature in the expression subgroups. (**B**) High expression of the signature correlates with an increased proportion of CMS4 tumors. (**C**) Meta-gene values of the liver metastasis signature expression in CMS1-4 subgroups. (**D**) Correlation plot of the expression of meta-gene values of the liver metastasis signature in relation to genes identifying CMS4 from the original random forest CMS classifier. Expression of (**E**) fibroblast activation protein (FAP), (**F**) smooth muscle actin (ACTA2), and (**G**) a signature reflecting the presence of tumor stroma (ESTIMATE stroma) in paired primary tumors and liver metastases.

**Table 1 cancers-16-01073-t001:** Comparing the metastatic efficiency values derived from the caecum and liver tumors. * spread to distant liver lobes other than the primary liver tumor. Generalized estimating equations (GEE) analysis.

From	To Liver * (*p* = 0.001)	To Lung (*p* = 0.017)	To Peritoneum (*p* = 0.01)
Caecum	40%	7%	33%
Liver	51%	30%	51%

## Data Availability

The authors declare that the data supporting the findings of this study are available within the paper and its Appendix A. The material and sequencing data supporting the findings of this study are available from the corresponding author upon reasonable request, according to the data protection assessment impact as part of the University Medical Center Utrecht (UMCU) FAIR principle.

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
