# Peer review of "Onward Spread from Liver Metastases Is a Major Cause of Multi-Organ Metastasis in a Mouse Model of Metastatic Colon Cancer"

_cancers, 2024, doi:10.3390/cancers16051073_

Round 1

Reviewer 1 Report

Comments and Suggestions for Authors

1. The author needs to clearly explain the detailed method of tumorigenesis in gene model mice, as well as the sources of subsequent organoids? Is it a primary intestinal tumor or a liver tumor? The biological behavior of organoids produced by tumors in different parts of gene model mice varies greatly. The author needs to clearly express the source of the organoid used.

2. The author found that the tumorigenicity and migration ability of organoids in the primary site of the intestine and liver are different, with the microenvironment being a potential key factor. Besides differences in angiogenesis ability, do factors such as immune cell composition and metabolic environment also play a key role? Suggest improving single-cell sequencing to further clarify.

3. The author conclude that liver metastases are efficient instigators of multi-organ metastasis, and that liver-specific micro-environmental signals are likely to contribute to such ‘on-ward spread. To demonstrate this, it is necessary to compare the differences in metastasis ability among organoid tumor models in different parts of the liver, lungs, and peritoneum.

4.Lack of exploration of molecular mechanisms in overall research. The research results are difficult to fully explain the author's conclusion.

Comments on the Quality of English Language

The language of the article needs further polishing

Reviewer 2 Report

Comments and Suggestions for Authors

It's well known that colorectal cancer has multiple ways to metastasize and understant biologically properties of these different clinical presentations it's challenging. Not only organ invasion is important but also the type of invasion in the same organ has different prognosis and potential different biological characteristics. For instance, patients with only small and less than four liver metastases has extremely better prognosis compared with patients with extensive liver metastases and high lactate dehydrogenase levels and patients with low peritoneal spread (ICP<11) has better prognosis than patients with ICP>21. The study presented by Wijler and cols, has several flaws that limit the interest for the readers

They conclude that colon tumors in the liver seeded more efficiently than those seeded in the primary tumor, but the four organoids come from the same mouse intestinal cancer with Notch1 and TP53 deletion. It is really difficult to recapitulate the complexity colorectal cancer (mutations, transcriptomics and pots-translational) with a unique murine model. In addition, other typical murine colorectal cancer models (MC38 and CT26) have shown the difficulty to translate to the clinical scenario.

Reviewer 3 Report

Comments and Suggestions for Authors

Dear Authors,

The aim of your original work „was to gain insight into the processes governing multi-organ metastasis in microsurgical transplantation of mouse colon tumor organoids into the caecum or into the liver of syngeneic immunocompetent mice”.

I have carefully reviewed your manuscript. Overall, I found this topic very interesting and relevant to the field of colon and/or colorectal cancer (CRC) research. The topic is important, especially because the mechanisms of multi-organ metastasis in this type of cancer are incompletely understood.

The objectives and the introduction of the thesis are written clearly, I have no comments on this part of the thesis. Good research hypotheses were set. And efforts were made to solve them gradually.

Material and Methods: A modern research model was used. The description of organoid culture, animals treatment and microsurgical implantation of tumor organoids into the caecum and the liver, bioluminescence imaging (BLI), immunohistochemistry (IHC), fluorescent multiplex IHC, whole genome sequencing and statistical analyses is correct. I have no comments on this description.

The Results are generally presented very clearly, and I have no major complaints, but I would recommend improving the quality of all the color images, as they are out-of-focus, especially the tissue results (Fig. 1B and C; all Fig. 2; Fig. 3B and C, Fig. 5). So, the quality of all attached figures and charts needs improvement. Perhaps this is a technical issue and you just need to save these figures in another graphics program.

In addition, Figure 1D is missing, and it is cited in the text of the paper (line 240).

The descriptions under the figures are correct and legible.

The discussion, though brief, is sensible, and written in a factual manner. The conclusions of the paper are also appropriate to the interesting work results obtained.

In general, I would also like to ask whether the mechanisms shown apply only to colon cancer or also to rectal cancer? Possibly add some commentary in the discussion, if it matters. 

References: Please check all the literature to make sure that everything is in accordance with the requirements of the editors, in paper 28 the title of the publication should be completed. And in paper No. 20, there are too many words "advance on, ", please correct.

To sum up:

This original paper provides valuable insights into mechanisms of „onward spread” from liver metastases in case of the colon cancer. The results are interesting and may explain, among others, why anti-angiogenic therapy yields better results when treating metastatic CRC, but not when it is part of adjuvant chemotherapy after resection of the primary tumor.

The paper is important from clinical point of view and may carry practical importance. Congratulations for all authors!!!

The paper can be accepted after minor revision.

Round 2

Reviewer 2 Report

Comments and Suggestions for Authors

The first revision identify critical flaws that there have been not rectified in the current version.